# VEC2PIX: CONTROLLABLE IMAGE SYNTHESIS VIA SEMANTIC-ALIGNED VECTOR GRAPHICS

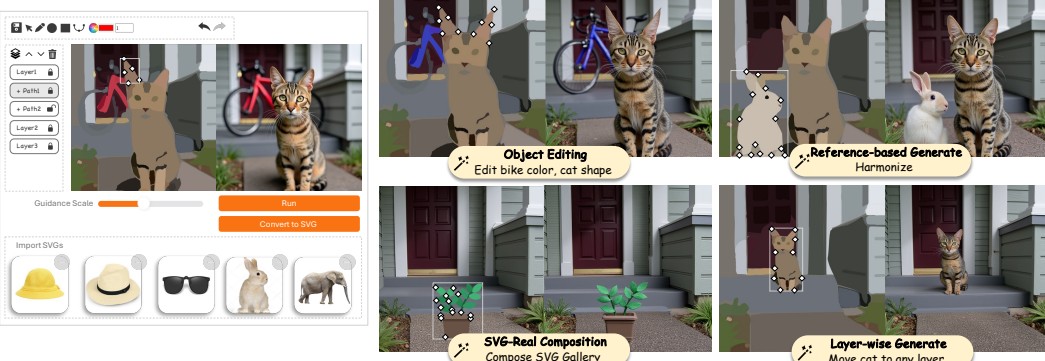

Figure 1: Visual examples of applications of the proposed **Vec2Pix** framework, where the left column depicts input SVGs and the right column presents the corresponding generated images. Vec2Pix offers 1) *Easy-to-control*: it supports layer-wise object insertion, removal, modification, color adjustment, shape editing, and flexible composition; 2) *High fidelity*: it caches semantic and color information through hierarchical SVG representations; and 3) *Strong input-generation alignment*: it ensures precise semantic and structure alignment between SVG inputs and generated images.

## ABSTRACT

Recent advances in image generation have achieved remarkable visual quality, while a fundamental challenge remains: Can image generation be controlled at the element level, enabling intuitive modifications such as adjusting shapes, altering colors, or adding and removing objects? In this work, we address this challenge by introducing layer-wise controllable generation through simplified vector graphics (VGs). Our approach first efficiently parses images into hierarchical VG representations that are highly semantic-aligned and structurally coherent. Building on this representation, we design a novel image synthesis framework guided by VGs, allowing users to freely modify elements and seamlessly translate these edits into photorealistic outputs. By leveraging the structural and semantic features of VGs in conjunction with noise prediction, our method provides precise control over geometry, color, and object semantics. Extensive experiments demonstrate the effectiveness of our approach in diverse applications, including image editing, object-level manipulation, and fine-grained content creation, establishing a new paradigm for controllable image generation. Project page: https://vec2pix-spec.github.io/Vec2Pix/

## 1 INTRODUCTION

Recent progress in image generation (Rombach et al., 2022c; Ho et al., 2020; Peebles & Xie, 2023; Labs, 2024), has led to remarkable advances in visual quality. However, in real-world scenarios, the results often fail to fully meet user expectations: local details may be unsatisfactory, and fine-grained controllability is lacking, making it difficult to flexibly edit specific regions or manipulate individual elements. Tasks such as adjusting an object's shape, changing its color, or adding or removing elements usually require complex prompt engineering or specialized editing pipelines, which can introduce artifacts and disrupt background consistency.

To address controllability, prior works explore conditional guidance such as sketches (Chen et al., 2009; Gao et al., 2021), layouts (Zhao et al., 2019; Sun et al., 2021), or drag-based interactions (Chen et al., 2023). These methods have benefited digital art and design, yet their operability remains limited — controls are either too coarse (e.g., layout) or too localized (e.g., drag), and often lack semantic awareness of the edited elements thus not easy to edit shape and attributes. This gap raises a fundamental question: *can image generation be controlled at the element level, enabling intuitive modifications such as adjusting shapes or sizes, altering colors, or adding and removing objects?*

In this work, we address this challenge by introducing a novel element-wise controllable generation framework by involving semantically-aligned vector graphics (VGs) as representations. Our key idea is to parse images into hierarchical vector representations, where the reconstructed vector graphics are both semantically aligned and structurally coherent. To this end, we propose an efficient vectorization pipeline that achieves over a $7\times$ speedup while producing VGs that are easy to modify. Building on this representation, we develop a generation framework guided by VGs, which enables users to freely modify elements and seamlessly translate these edits into photorealistic outputs. In particular, a tunable encoder predicts the initial noise from VGs, further aligning structural and semantic features for controllable synthesis. Extensive experiments demonstrate that our approach enables flexible and reliable element-wise editing across diverse applications, including image editing, object-level manipulation, realistic and vector graphics composition, and localized de-artifacts. We believe this establishes a new paradigm for controllable image generation, bridging the gap between high-quality synthesis and practical usability. Our contributions can be summarized as follows:

- We propose a controllable generation framework that leverages *Simplified Vector Graphics*, a hierarchical vector parsing of images that is semantically aligned and structurally coherent, offering interpretable and manipulable latents for user interaction.

- We present *Vector-Guided Noise Prediction*, where a tunable encoder derives the spatially variant initial noise of diffusion models from vector graphics, thereby aligning structural and semantic features for controllable synthesis.

- Experimental results show that our framework supports element-wise re-generation and modification, and significantly improves controllable image generation, enabling layout-wise generation, object-level editing, and composition.

## 2 PRELIMINARY

**Flow models** (Liu et al., 2023; Lipman et al., 2023; De Bortoli, 2022) parameterize the velocity field $u_t \in \mathbb{R}^d$. Recent advances such as Flux (Labs, 2024) apply this principle to text-to-image generation, where an image $I$ is encoded into a latent variable $z_0 = \text{Enc}(I)$ by a Variational autoencoder (VAE) encoder $\text{Enc}(\cdot)$ consisting of convolutional downsampling layers, residual blocks, and a mid-level attention mechanism. A symmetric decoder reconstructs an image $\hat{I}$ from a latent representation $\hat{z}_0$. The latent dynamics are modeled by a transformer-based velocity network, adopting a DiT (Peebles & Xie, 2023)-style design where latent patches are treated as tokens.

The flow model construction assumes a Gaussian prior $z_1 \sim \mathcal{N}(0, I)$ and connects it to the data latent $z_0$ through a linear interpolation path:

$$z_t = (1-t)z_1 + tz_0, \quad t \in [0,1], \tag{1}$$

with instantaneous velocity $u_t(z_t) = z_0 - z_1$.

The transformer denoiser $v_\theta$ is trained to approximate this velocity by minimizing the squared error:

$$\mathcal{L}_{\text{FM}}(\theta) = \mathbb{E}_{\boldsymbol{x}_0, \boldsymbol{z}_0, \boldsymbol{z}_1, t} \left[ |v_\theta(\boldsymbol{z}_t, t, c) - (\boldsymbol{z}_0 - \boldsymbol{z}_1)|_2^2 \right]. \tag{2}$$

Sampling starts from Gaussian noise $z_1$ by solving the ODE $\frac{dz_t}{dt} = v_\theta(z_t, t, c)$ from $t : 1 \to 0$.

**Vector graphics (VG)** represent images not as dense pixels but as geometric primitives such as points, lines, and curves. Among different primitives, *Bézier curves* are particularly popular for their ability to model smooth boundaries. Most existing image vectorization methods represent an image as a collection of *closed regions*, each described by an ordered sequence of Bézier curve segments:

$$\Omega = \{B_{i_1}, \ldots, B_{i_L}\}, \quad B_{i_\ell}(1) = B_{i_{\ell+1}}(0), \ B_{i_L}(1) = B_{i_1}(0), \tag{3}$$

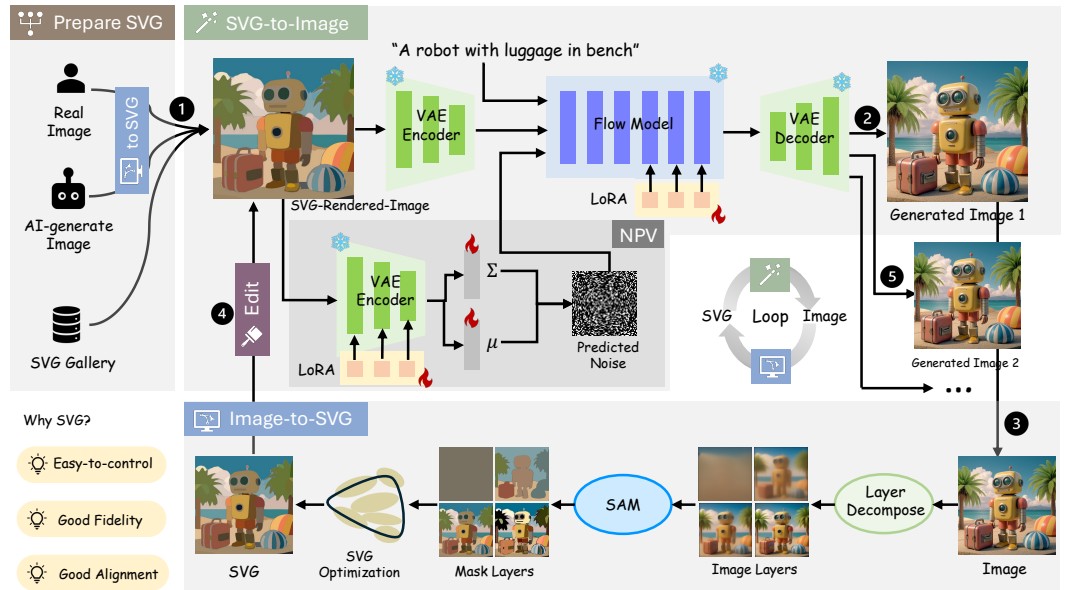

Figure 2: Overall framework of our Vec2Pix and its workflow. ① **Prepare SVG:** the input is obtained by converting a real or AI-generated image into SVG, or by selecting an existing SVG from a gallery. ② **SVG-to-Image:** the SVG information will be conditioned using token concatenation and noise prediction from vectors (NPV) module. The NPV module incorporates the SVG condition and integrates trainable LoRA adapters and prediction heads to estimate the mean and variance of the initial noise, rather than directly sampling from Gaussian noise. *If the user wishes to re-generate or modify specific parts, we proceed with steps* ③–⑤. ③ **Image-to-SVG:** the generated image is converted back into SVG using a diffusion model to produce multiple layers, followed by SAM to generate semantic masks for each layer, and further refined via 2D Gaussian optimization. ④ **SVG Editing:** users can interactively edit the SVG by adjusting curves and attributes. ⑤ **Re-generation:** the modified SVG is used as guidance to synthesize the final updated result.

where $\Omega$ denotes the closed region, and each $B_i$ is a Bézier segment. The segments are ordered so that the endpoint of $B_{i_\ell}$ matches the start of $B_{i_{\ell+1}}$, and the last segment $B_{i_L}$ connects back to the first $B_{i_1}$. Thus, the index $i$ specifies the sequence of segments, ensuring that all $B_i$ are linked head-to-tail to form a continuous closed curve.

A Bézier segment $B_i(t)$ of degree $M$ is parameterized as

$$B_i(t) = \sum_{j=0}^{M} \binom{M}{j}(1-t)^{M-j}t^j P_j^{(i)}, \quad t \in [0,1], \ P_j^{(i)} \in \mathbb{R}^2. \tag{4}$$

Here, $M$ denotes the polynomial degree, and $P_j^{(i)}$ are the control points. In our paper, we used *cubic Bézier curves* ($M = 3$), as they strike a balance between expressiveness and simplicity.

The emergence of *differentiable vector graphics rendering* (DiffVG (Li et al., 2020) and Bézier Splatting (Liu et al., 2025)) makes it possible to propagate gradients through rasterization. This allows the control points to be treated as *optimizable parameters*, refined by minimizing the discrepancy between the rasterized result and the target image. A common objective is the pixel-wise reconstruction loss:

$$\mathcal{L}_{recon} = \left\| I_{\text{real}} - R(\{\Omega_k\}) \right\|_1, \tag{5}$$

where $R$ denotes the differentiable rasterizer applied to the set of regions $\{\Omega_k\}$. This formulation establishes a direct link between continuous vector representations and gradient-based optimization, providing the foundation for trainable and editable vector graphics.

# 3 METHODOLOGY

## 3.1 OVERALL FRAMEWORK

Unlike prior controllable image generation approaches based on sketches, layouts, or depth maps (Gao et al., 2021; Zhao et al., 2019; Tan et al., 2025), this work adopts **vector graphics**, like SVG, as the conditioning representation, offering a more flexible, structured, and easy-to-edit description of visual content. Vector graphics inherently capture the hierarchical relationships between foreground objects and background elements within a scene, enabling richer control and composition. This structure makes them more intuitive for user manipulation, enabling direct editing and control over individual components. In addition, vector graphics seamlessly integrate with existing SVG libraries, allowing easy reuse and composition of existing rich design assets.

To this end, we design a loop of **SVG⇌image** consisting of two key components: (1) an *SVG-guided image generation module* that ensures the synthesized image faithfully follows both user-manipulated vector graphics and the accompanying text prompt (see Section 3.2), and (2) an *image-to-SVG parsing module* that converts images back into vector graphics with hierarchical semantic representations (see Section 3.3). Together, these components form a closed loop that enables element-wise controllable generation and iterative refinement through user interactions. Formally, the loop can be expressed as

$$\mathcal{S} \xrightarrow{G_{\text{SVG}\to\text{Img}}} \hat{\mathcal{I}} \xrightarrow{P_{\text{Img}\to\text{SVG}}} \hat{\mathcal{S}} \xrightarrow{\text{edit}} \hat{\mathcal{S}}' \xrightarrow{G_{\text{SVG}\to\text{Img}}} \hat{\mathcal{I}}', \tag{6}$$

where $\mathcal{S}$ denotes the input SVG representation, $\hat{\mathcal{I}}$ is the generated image, $\hat{\mathcal{S}}$ is the reconstructed SVG, $\hat{\mathcal{S}}'$ is the user-modified SVG, and $\hat{\mathcal{I}}'$ is the re-generated image.

## 3.2 EFFICIENT AND SEMANTIC-ALIGNED VECTOR GRAPHICS

Unlike existing image vectorization approaches (Adobe Inc., 2025; Li et al., 2020; Ma et al., 2022), which mainly focus on refining fine-grained details but often lack semantic meaning, offer limited editability, and are inefficient, we propose a new Image-to-SVG parsing module that delivers a user-friendly vectorized representation through *efficient, layer-wise, and semantically aligned decomposition*, enabling intuitive element-level adjustments.

We build on the layer decomposition strategy of LIVSS (Wang et al., 2025), leveraging visual foundation models (*e.g.*, Segment Anything (Kirillov et al., 2023) and Stable Diffusion (Rombach et al., 2022b)) to introduce semantic priors, thereby producing vector graphics that are both layer-wise and semantically aligned. For each layer, we adopt the efficient differentiable renderer, *Bézier Splatting* (Liu et al., 2025), which splats Gaussian kernels along curves and yields stable gradients at over an order-of-magnitude lower computational cost. To further ensure compactness, each semantic region is constrained to at most 12 cubic Bézier segments. Together, these designs produce vector graphics that are semantically aligned, structurally simple, easy to edit, and computationally efficient. In the following, we discuss more details of our vector graphics representation.

**Layer-wise and semantic-aligned vector graphics initialization.** Following LIVSS, diffusion-based smoothing generates a sequence of progressively simplified images $\{I_S^{(t)}\}_{t=1}^S$, where $S$ denotes the number of simplified images. Each image $I_S^{(t)}$ is segmented by SAM into semantic masks $\{I_{\text{mask}}^{(t)}\}_{m=1}^{M_t}$. The raw masks are not directly used; instead, they are filtered and organized into layers based on the order of the simplified images, the relative mask size, and the degree of overlap with existing masks. Redundant masks are removed, and the remaining ones are assigned to layers such that coarser masks from heavily smoothed images are placed in deeper layers, while finer non-overlapping masks from less-smoothed images populate shallower layers. This process yields a layered collection $\mathcal{M} = \{\mathcal{M}^{(1)}, \dots, \mathcal{M}^{(L)}\}$, where $L$ denotes the total number of layers. Finally, to initialize vector graphics $\{\Omega_k\}$, mask boundaries are polygonized, simplified with the Douglas–Peucker algorithm (Douglas & Peucker, 1973), and curve-fitted. However, irregular masks often lead to overly complex shapes with dense control points, making the resulting SVGs difficult to edit. To address this limitation, we adopt a more compact initialization scheme. For each semantic mask with a *simplified polygon*, we identify the two vertices with the longest diagonal distance and use them as splitting points. The polygon is then divided into two sub-chains, each approximated with a fixed number of cubic Bézier segments. Although this produces coarser initial boundaries than

LIVSS, our representation remains fully differentiable, allowing subsequent optimization to refine the curves into accurate shapes. This design ensures that initialization complexity is strictly bounded while preserving the ability to converge to precise object contours.

**Efficient differentiable vector graphics rasterization.** Building upon the layered initialization, we employ the Bézier Splatting (Liu et al., 2025) for vector graphics rasterization, which splats differentiable 2D Gaussian (Zhang et al., 2024) along curve trajectories to calculate the boundary gradient efficiently. A challenge arises because Bézier Splatting computes pixel colors through alpha blending, which assumes that each pixel is formed by accumulated transparent primitives. This assumption is incompatible with layer-wise mask optimization, where each pixel should belong exclusively to a single semantic region. As a result, when directly supervised by binary masks, Bézier Splatting may reduce loss by expanding regions and adjusting opacities, leading to lower PSNR and optimization trapped in poor local minima. To eliminate this degeneracy, we fix the opacity of all regions to 1.0, ensuring that masks remain opaque and that gradients drive boundary alignment rather than opacity adjustment. This modification makes Bézier Splatting stable and effective for layer-wise semantic vectorization.

To optimize the initialized vector graphics $\{\Omega_k\}$ while preserving their layer-wise semantic alignment, we employ a complementary losses. The structure loss supervises geometry by matching each rendered region $I_{\text{vec}}^k$ with its corresponding semantic mask $I_{\text{mask}}^k$:

$$\mathcal{L}_{\text{structure}} = \sum_{k=1}^{|\mathcal{K}|} \|I_{\text{mask}}^k - I_{\text{vec}}^k\|. \tag{7}$$

The final training objective is:

$$\mathcal{L} = \mathcal{L}_{\text{structure}} + \gamma \mathcal{L}_{\text{recon}}, \tag{8}$$

where $\mathcal{L}_{\text{recon}}$ refers to Eq. 5 and $\gamma$ is the loss weight. It enforces semantic alignment of boundaries and photometric fidelity of appearance, resulting in compact and easily editable SVGs .

### 3.3 VECTOR-GUIDED CONTROLLABLE IMAGE GENERATION

We adopt a two-stage training strategy for the SVG-guided image generation process $G_{\text{SVG} \rightarrow \text{Img}}$. In the **first stage**, we integrate LoRA modules into transformer blocks to adapt the original text-to-image model, *i.e.*, Flux.1-dev (Labs, 2024), to text and SVG conditioned generation model. Specifically, the image branch encodes the parsed SVG into feature representations, which are concatenated with noise along the channel dimension to initialize generation. In parallel, the text branch encodes the accompanying description to provide semantic guidance. These two branches are fused through multimodal attention, allowing the model to jointly attend to visual and textual cues (Tan et al., 2025). We formulate this integration as:

$$Q = [Q_t; Q_z; Q_c], \quad K = [K_t; K_z; K_c], \quad V = [V_t; V_z; V_c], \tag{9}$$

$$\text{MMA}(Q, K, V) = \text{softmax}\left(\frac{QK^T}{\sqrt{d}}\right) V. \tag{10}$$

where $[;]$ represents the concatenation operation, and $Q$, $K$, and $V$ are the query, key, and value components of the attention mechanism.

We introduce **Noise Prediction from Vectors (NPV)** as the **second stage** to further strengthen the structural alignment between SVGs and generated images. Specifically, we add a noise prediction module that estimates the mean and variance of the initial noise of flow matching model, such that the guidance from SVGs is directly injected at the beginning phase of the generation process. We extend the Flux model's VAE encoder with low-rank adapters (LoRA) and lightweight head tuning to obtain the latent distribution parameters. Given an input image $x$, the encoder produces the mean $\mu$ and the log-variance $\log \sigma^2$ as

$$\mu, \log \sigma^2 = \left(W_\mu, W_{\log \sigma^2}\right) * \phi\left(\text{Enc}_{\widetilde{\theta}}(I)\right), \tag{11}$$

where $\text{Enc}_{\widetilde{\theta}}(\cdot)$ denotes the frozen Flux VAE encoder plugged with learnable LoRA modules, where the parameters are modified as $\widetilde{\theta} = \theta^{(0)} \cup \{\Delta W_l\}_l$ with LoRA updates injected into selected layers. Here, $\phi$ represents a GroupNorm layer followed by a SiLU activation, and $W_\mu, W_{\log \sigma^2}$ are parallel $1 \times 1$ convolutional heads predicting $\mu$ and $\log \sigma^2$, respectively.

Figure 3: Our Vec2Pix supports various controllable image generation and editing tasks.

With these parameters, we sample the latent code $z$ via the reparameterization trick:

$$z = \mu + \sigma \odot \epsilon, \quad \epsilon \sim \mathcal{N}(0, I). \tag{12}$$

The training objective consists of three components: the flow matching loss $\mathcal{L}_{\text{FM}}$ from Eq. 2, the KL loss $\mathcal{L}_{\text{KL}}$ to minimize the divergence from the standard normal distribution, and a covariance loss $\mathcal{L}_{\text{cov}}$ to encourage spatially independence across latent channels:

$$\mathcal{L}_{\text{KL}} = \frac{1}{2} \mathbb{E}\Big[\mu^2 + \sigma^2 - \log \sigma^2 - 1\Big]. \tag{13}$$

$$\mathcal{L}_{\text{cov}} = \frac{1}{C(C-1)} \sum_{i \neq j} R_{ij}^2, \qquad R = \frac{\tilde{\mu}\,\tilde{\mu}^\top}{\|\tilde{\mu}\|^2}, \tag{14}$$

where $\tilde{\mu}$ is the channel-normalized mean vector, and $C$ is the number of elements in the window. To avoid the large memory overhead and artifacts caused by computing covariance with a global fixed window, we randomly select $N$ patches of size $p \times p$ and compute the mean covariance loss over the patches. The final objective is a weighted combination, as:

$$\mathcal{L} = \mathcal{L}_{\text{FM}} + \beta\,\mathcal{L}_{\text{KL}} + \lambda\,\mathcal{L}_{\text{cov}}. \tag{15}$$

## 4 EXPERIMENTS

### 4.1 EXPERIMENTAL SETUPS

**Implementation details.** Our Vec2Pix builds on FLUX.1-dev (Labs, 2024), a latent rectified flow transformer for text-to-image generation. We apply LoRA (Hu et al., 2021) to the transformer blocks of the base model, using a default rank of $4$, with the LoRA scale set to 0 by default for non-condition tokens, introducing approximately 14.5M trainable parameters for stage one. In addition, we apply LoRA with rank of $8$ to the Flux VAE encoder at both the down and middle stages, introducing approximately 135K trainable parameters to predict the initial noise. Our model is trained with a batch size of $1$ and gradient accumulation over $4$ steps. We employ the Prodigy optimizer with safe-guard warmup and bias correction enabled, setting the weight decay to $0.01$. We set hyperparameter $\gamma = \beta = \lambda = 1$ and the number of sampled patches as $N = 8192$ and patch size $p = 4$. All experiments are conducted on 4 NVIDIA A100 GPUs (80GB each). For stage one, our model is trained for 50K iterations, while stage two is trained for 10K iterations.

**Dataset construction.** We filter around 5M images and their corresponding text prompts from the LAION-400M dataset, retaining only those with resolutions larger than $1024 \times 1024$. All training data are resized to $512 \times 512$ for training. We then employ our proposed efficient Image-to-SVG module to convert these images into SVGs, thereby constructing {image, SVG, text} triplets for the training set. We adopt this dataset consistently for both stage one and stage two training processes.

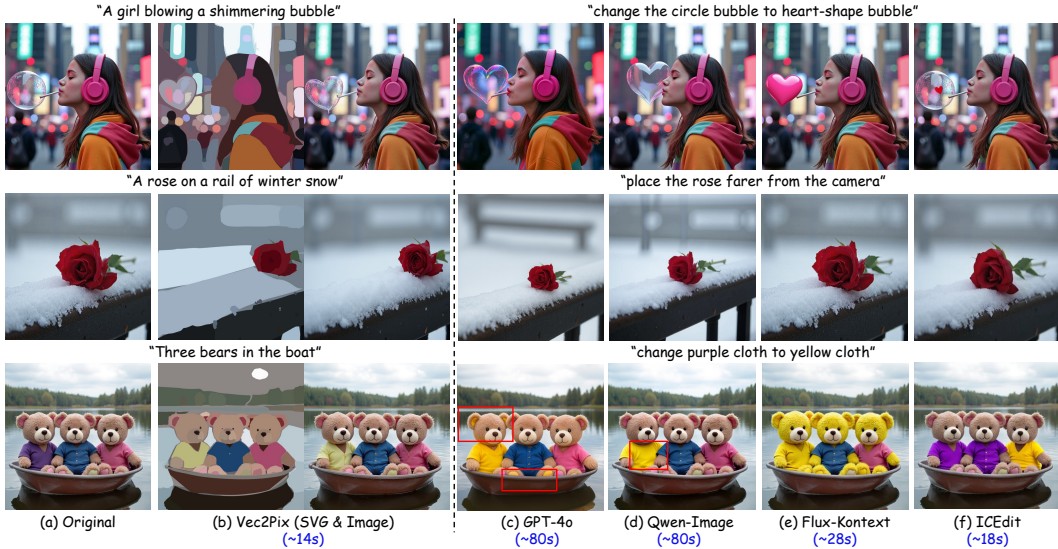

"A girl blowing a shimmering bubble"  "change the circle bubble to heart-shape bubble"

"A rose on a rail of winter snow"  "place the rose farer from the camera"

"Three bears in the boat"  "change purple cloth to yellow cloth"

(a) Original  (b) Vec2Pix (SVG & Image) (~14s)  (c) GPT-4o (~80s)  (d) Qwen-Image (~80s)  (e) Flux-Kontext (~28s)  (f) ICEdit (~18s)

Figure 4: Visual comparisons with text-prompt-guided editing methods, including state-of-the-art open-source and commercial solutions such as GPT-4o , Qwen-Image Wu et al. (2025), Flux-Kontext Labs et al. (2025), and ICEdit Zhang et al. (2025). Editing cases such as shape modification, object repositioning, and color adjustment are readily supported by our VG representation, whereas text-guided editing often fails.

## 4.2 APPLICATIONS

Figure 3 showcases the applications supported by our Vec2Pix method, while Figure 4 compares its editing performance with recent state-of-the-art approaches. Vec2Pix enables a wide range of re-generation and editing tasks, including:

**Layer-wise generation.** Vec2Pix enables controllable layer-wise generation, where different semantic layers (*e.g.*, background, mid-level structures, and foreground details) can be synthesized independently. For example, the object can be positioned at any layer, such as placing the building behind existing trees, inserting furniture within the mid-level interior space. This design enables flexible scene composition while keeping local edits semantically consistent and visually faithful, through SVG that preserves semantic features via hierarchical layering.

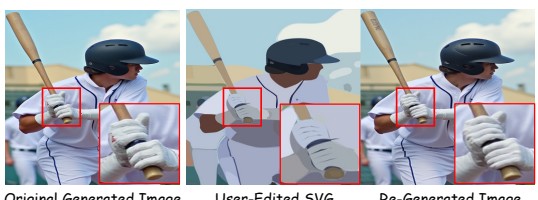

Original Generated Image  User-Edited SVG  Re-Generated Image

Figure 5: Our Vec2Pix can be applied to artifact removal. Fine-grained elements in the original generation, such as fingers, are sometimes rendered incorrectly (*e.g.*, **three fingers** instead of four in a frontal view). By applying manual corrections or guidance in SVG, the re-generated image can be effectively refined to eliminate such artifacts.

**Object editing.** Vec2Pix supports object editing, allowing precise manipulation of individual components within a scene. Users can alter object attributes such as shape, color, or position, which is especially beneficial for design and creative applications. For instance, replacing a dog with a cat and rabbit, changing the color of a bench, or modifying the shape of mountain. These edits preserve contextual consistency, keeping the scene semantically coherent and visually realistic.

**Reference-based generation.** Vec2Pix enables reference-guided synthesis, in which objects' structures and color can be transferred from one or multiple exemplars, in the mean time, the SVG controls structures. Then generated harmonized results in the scene such as the inserted cat, rabbit, and elephant in Figure 3(b). Vec2Pix can mitigate background leakage and resolves conflicting cues via learned priors, yielding photorealistic results faithful to both the references and the SVG layout.

**Real & SVG composition.** By leveraging vector-based SVG representations, our approach enables flexible composition of SVGs and real images within a single complex scene. Vector primitives

Table 1: Quantitative comparison with controllable generation baselines across different representation types including sketch, depth, stroke, segmentation mask, and our proposed SVG, along with our variants trained with and without noise prediction from vectors (NPV). We report both reconstruction and editing quality, with the best results highlighted in **bold**.

| Representation | Base Model | Reconstruction | | | | Editing | | | |
|---|---|---|---|---|---|---|---|---|---|
| | | FID↓ | LPIPS↓ | SSIM↑ | PSNR↑ | FID↓ | LPIPS↓ | SSIM↑ | PSNR↑ |
| Canny | FLUX.1 | 17.38 | 0.4582 | 0.42 | 12.28 | 22.48 | 0.5639 | 0.37 | 11.54 |
| Depth | FLUX.1 | 19.65 | 0.5572 | 0.36 | 11.60 | 22.94 | 0.5932 | 0.36 | 11.37 |
| Stroke | FLUX.1 | 22.42 | 0.4721 | 0.45 | 17.13 | 25.35 | 0.5249 | 0.44 | 14.87 |
| Segmentation Mask | FLUX.1 | 17.75 | 0.4513 | 0.43 | 15.78 | 21.97 | 0.5382 | 0.43 | 14.34 |
| Ours (SVG) w/o NPV | FLUX.1 | **16.03** | **0.3687** | **0.47** | **17.42** | **18.87** | **0.3901** | **0.47** | **17.25** |
| Ours (SVG) w NPV | FLUX.1 | **15.52** | **0.3515** | **0.49** | **17.77** | **17.84** | **0.3836** | **0.50** | **17.72** |

can be rearranged, combined, or modified to form realistic scenes such as the cat in a bookstore shown in Figure 3(a), and also allow the insertion of realistic objects (e.g., the man) into SVG-based conditions.

**Localized de-artifacts.** Vec2Pix can also address visual artifacts. When fine-grained details in the initial generation are flawed, such as fingers, are sometimes rendered incorrectly (*e.g.*, wrong number of fingers), manual adjustments or SVG-based guidance enable the re-generated image to be refined and corrected as shown in Figure 5.

### 4.3 METHOD COMPARISON

To evaluate the representation ability and controllability of Vec2Pix, we construct an evaluation set by collecting 5K paired source–target images from existing datasets (Yu et al., 2024). Each source image is converted into four structural conditions, including Canny edges, depth maps, strokes (Meng et al., 2022), Segmentation Mask Kirillov et al. (2023), and our proposed SVG representation, using publicly available methods. We then perform conditional generation to reconstruct the source image from each condition. Specifically, we adopt the well-trained Canny- and depth-conditioned models released in OmniControl (Tan et al., 2025), while the stroke and segmentation mask-conditioned model are trained with the same data as our SVG-conditioned model. All methods, including ours, are built upon the FLUX.1-dev base model. For evaluation, we measure reconstruction quality with FID (Heusel et al., 2017), PSNR (Horé & Ziou, 2010), SSIM (Wang et al., 2004), and LPIPS (Zhang et al., 2018). Beyond reconstruction quality, we assess controllability by examining the editability of different representations. Given source–target pairs, we compute modification masks and automatically apply them to the representations (*e.g.*, $\text{mask} \times \text{target}_{\text{SVG}} + (1 - \text{mask}) \times \text{source}_{\text{SVG}}$). The edited representations are then used for conditional generation, and the resulting images are compared against the target images to quantify editing performance. Table 1 summarizes the reconstruction and editing results, demonstrating that our SVG serves as an effective representation for both information caching and editability.

### 4.4 ABLATION STUDY

**The effect of noise prediction.** In stage one, training treats SVGs as conditional inputs via token concatenation. Stage two adds noise prediction through the NPV module, further aligning SVG structure with generated outputs. As shown in Table 1 compares reconstruction quality with and without NPV variances, showing that incorporating NPV leads to better alignment in the generated results. Besides, as shown in Fig. 6(a), incorporating NPV makes the structure of the generated results align more closely with the input SVG, for example, the mountain silhouette. Importantly, the mountain's reflection in the generated image still obeys symmetry about the waterline, as encoded by the base model's physics-consistent prior, despite a non-physical input SVG (*e.g.*, asymmetric reflection edges). With more stage-two training iterations, adherence to the input structure strengthens, sometimes forcing physically implausible reflections.

**Effect of vector scale adjustment.** We adjust the strength of the SVG condition by varying a scaling factor, where we plot the PSNR and FID metrics under different scaling values, demonstrating how the factor influences both reconstruction quality and perceptual fidelity as shown in Figure 6(b).

**Effect of efficient and semantic-aligned vector graphics.** To assess the effectiveness of our efficient and semantic-aligned image-to-SVG module, we compare it with the previous semantic-

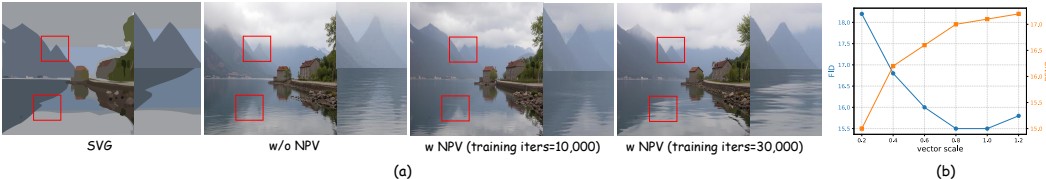

SVG      w/o NPV      w NPV (training iters=10,000)      w NPV (training iters=30,000)

(a)                                                      (b)

Figure 6: (a) Ablation study comparing results with and without the proposed Noise Prediction from Vectors (NPV) module, as well as training NPV module with different iterations. (b) PSNR and FID performance variations under different vector scales used to adjust the conditioning strength.

Table 2: Comparison of reconstruction quality and efficiency between the existing semantic-layered image vectorization method LIVSS (Wang et al., 2025) and our proposed efficient, semantically simplified image vectorization module.

| Method | PSNR ↑ | Time (s) ↓ | Speedup |
|---|---|---|---|
| LIVSS (Wang et al., 2025) | 16.960 | 18.564 | 1.0× |
| **Our Image-to-SVG** | **17.353** | **2.656** | **7.0×** |

layered vectorization method LIVSS (Wang et al., 2025) on the DIV2K dataset (Agustsson & Timofte, 2017), which contains 1K high-quality images. All methods are optimized for 30 steps and resized to $512 \times 512$ resolution for a fair comparison. Table 2 reports results on both efficiency and reconstruction quality. The results demonstrate the advantages of our compact initialization and efficient Bézier Splatting rasterizer: under the same 30 optimization steps, our method achieves higher PSNR while being 7× faster.

## 5 RELATED WORK

**Image vectorization.** Image vectorization has advanced rapidly, but traditional tools like Illustrator's Image Trace are non-differentiable, often producing overly complex shapes that are difficult to edit. A key breakthrough came with DiffVG (Li et al., 2020), which pioneered differentiable rasterization and enabled gradient-based optimization of arbitrary Bézier curves, laying the foundation for compact and flexible vectorization. Building on this, LIVE (Ma et al., 2022) and O&R (Hirschorn et al., 2024) introduced layer-wise initialization strategies to generate more compact, topology-preserving representations, while LIVSS (Wang et al., 2025) further integrated SegmentAnything (SAM) (Kirillov et al., 2023) and diffusion priors to align SVGs with semantic content. Despite these advances, differentiable methods remain computationally expensive, often requiring hours for high-resolution optimization. Recently, Bézier Splatting (Liu et al., 2025) recasts rasterization as splatting, yielding significantly faster optimization while preserving Bézier-curve flexibility. In this work, we integrate Bézier Splatting with layer-wise semantic alignment to enable high-quality image vectorization in seconds.

**Controllable image generation.** Controllable generation in diffusion models has progressed from early text-to-image systems (Rombach et al., 2022a; Saharia et al., 2022) to spatially guided methods such as ControlNet (Zhang et al., 2023a); UniControl (Zhao et al., 2023) further unifies diverse spatial conditions under a Mixture-of-Experts (MoE) paradigm. However, because these approaches inject spatial condition features into the denoising hidden states, they are best suited to aligned inputs and struggle with misaligned or subject-driven generation. Extensions like IP-Adapter (Ye et al., 2023) (cross-attention with an auxiliary encoder) and SSR-Encoder (Cao et al., 2023) strengthen identity preservation, yet most methods still operate in pixel space (Qin et al., 2023; Ruiz et al., 2023; Shi et al., 2023; Tumanyan et al., 2023; Wang et al., 2023; Zhang et al., 2023b). In contrast, we move beyond purely spatial conditioning and explore *layer-wise controllable generation*, offering a more flexible and intuitive representation for fine-grained editing.

## 6 CONCLUSION

In this work, we have presented a new paradigm for controllable image generation based on semantic-aligned vector graphics. By decomposing images into hierarchical, semantically aligned vector representations and integrating them into a noise-guided generation framework, our method enables precise element-level control over geometry, color, and object semantics. The proposed approach not only delivers photorealistic outputs consistent with user edits but also supports a wide range of applications, from intuitive image editing to fine-grained object manipulation. These re-

sults highlight the potential of vector-guided generation as a foundation for the next generation of controllable and creative image synthesis.

## ETHICS STATEMENT

All experiments use publicly available datasets and model checkpoints (FLUX.1-dev). No human or animal subjects are involved. The method is intended for research only, and the authors declare no competing interests.

## REPRODUCIBILITY STATEMENT

All datasets are public, and implementation details, hyperparameters, and proofs are provided in the appendix. Code will be released to ensure full reproducibility.

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

# A    APPENDIX

## A.1    QUANTITIVE RESULTS OF OUR VEC2PIX FOR GENERATION AND EDITING

Figure 7: Examples of images generated and re-generated using our Vec2Pix framework.

## A.2 QUANTITATIVE RESULTS OF OUR EFFICIENT IMAGE-TO-VECTOR GRAPHICS ALGORITHMS

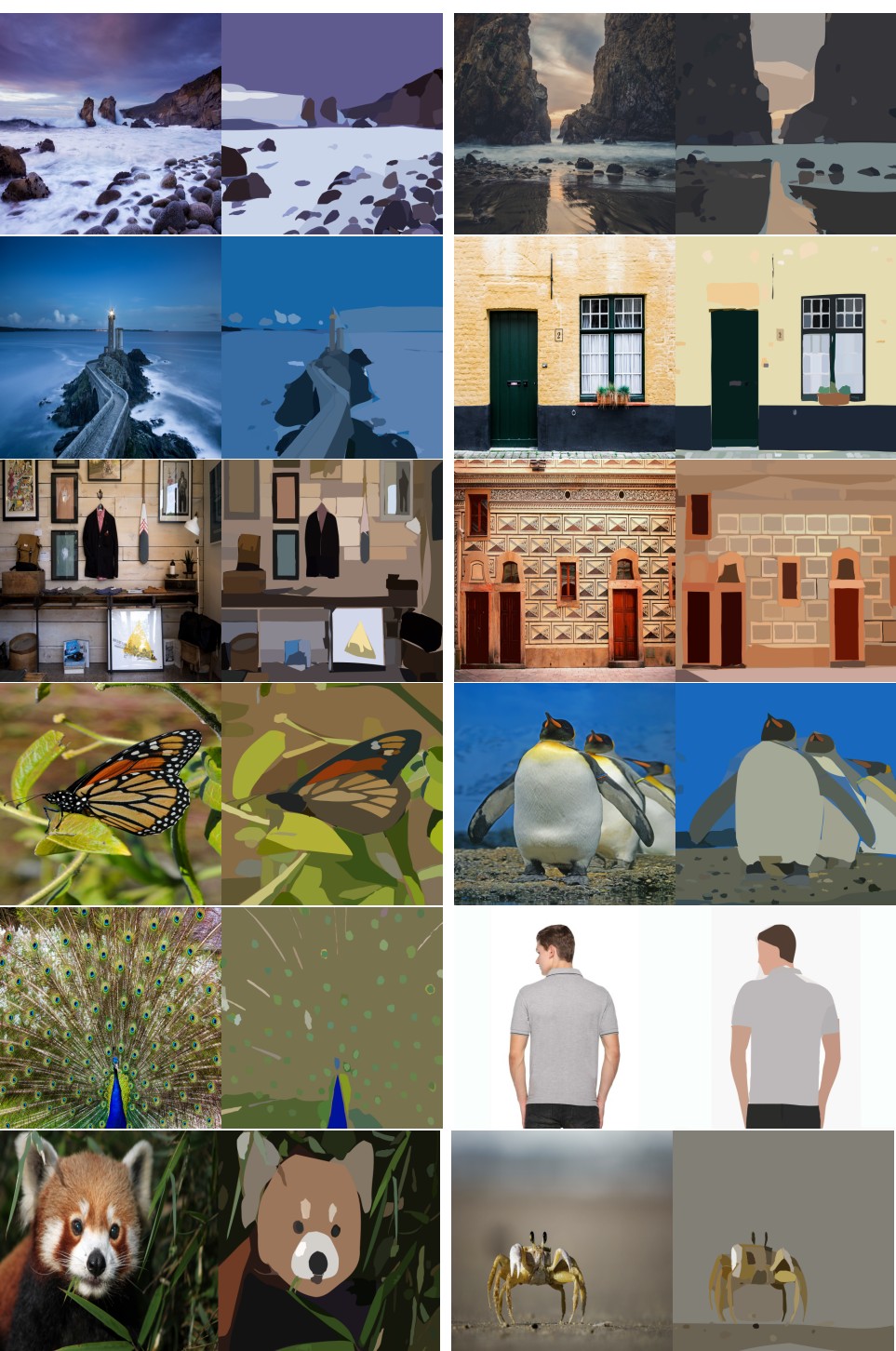

Figure 8: Real-world images in the first and third columns are sampled from the DIV2K Agustsson & Timofte (2017) dataset, while the second and fourth columns show their corresponding SVG conversions.

## A.3   MORE RESULTS OF NOISE PREDICTION FROM VECTOR

To evaluate the effectiveness of NPV, we also integrate it into other conditional generation settings (i.e., Canny and Stroke). The results in the table below (with and without NPV) show that NPV generally improves reconstruction quality across different conditioning types as shown in Table 3. We further apply NPV to a general editing method (i.e., Flux Knotext), where fidelity and generative diversity naturally form a trade-off in editing tasks, the resulting performance varies depending on the specific case as shown in Figure 9.

Table 3: Ablation studies on adding our proposed Noise Prediction from Vector (NPV) on various conditional-generation tasks, including Canny, Stroke, and our proposed SVG conditions.

| Representation | Base Model | Reconstruction | | | | Editing | | | |
|---|---|---|---|---|---|---|---|---|---|
| | | FID↓ | LPIPS↓ | SSIM↑ | PSNR↑ | FID↓ | LPIPS↓ | SSIM↑ | PSNR↑ |
| Canny | FLUX.1 | 17.38 | 0.4582 | 0.42 | 12.28 | 22.48 | 0.5639 | 0.37 | 11.54 |
| Canny w NPV | FLUX.1 | 16.89 | 0.4501 | 0.44 | 12.65 | 21.76 | 0.5573 | 0.41 | 12.32 |
| Stroke | FLUX.1 | 22.42 | 0.4721 | 0.45 | 17.13 | 25.35 | 0.5249 | 0.44 | 14.87 |
| Stroke w NPV | FLUX.1 | 21.88 | 0.4682 | 0.47 | 17.33 | 25.01 | 0.5204 | 0.46 | 15.13 |
| Ours (SVG) w/o NPV | FLUX.1 | **16.03** | **0.3687** | **0.47** | **17.42** | **18.87** | **0.3901** | **0.47** | **17.25** |
| Ours (SVG) w NPV | FLUX.1 | **15.52** | **0.3515** | **0.49** | **17.77** | **17.84** | **0.3836** | **0.50** | **17.72** |

**Text Prompt:**

"Change the cat into a dog"

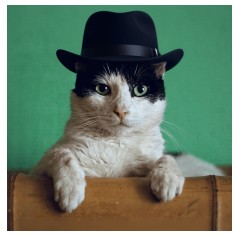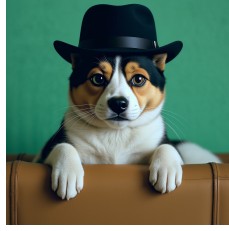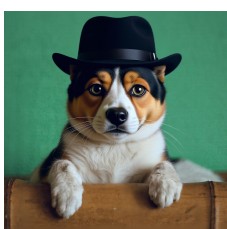

Input          Flux-Kontext          Flux-Kontext + NPV

Figure 9: A visual example of integrating our proposed Noise Prediction from Vector (NPV) module into a general editing method (i.e., Flux-Kontext). NPV typically **enhances fidelity** during the editing process. For example, the **texture of the table below and the shape and position of the animal's grasping hand are better preserved**.

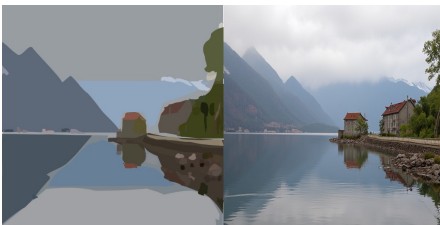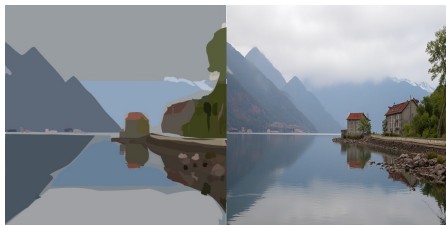

Guidance scale=1          Guidance scale=2

Figure 10: Our Vec2Pix framework allows flexible adjustment of the guidance scale to control how strongly the generation aligns with the input conditions. This scale determines whether the model should adhere closely to the SVG and allow precise object-level edits without altering shadows or reflections, or whether it should instead prioritize the physical realism captured by the base model. For example, users can control whether the mountain's reflectance follows its true geometric shape or follows the SVG guidance.

## A.4   VISUAL COMPARISONS ON VARIOUS GUIDANCE SCALE

We employ the guidance scale factor to control the alignment between the SVG conditions and the generated results. This factor also governs how the edited object interacts with its surrounding environment. Specifically, it determines whether the generation should adhere more strictly to the input SVG—enabling object-level edits without modifying shadows or reflections—or whether it

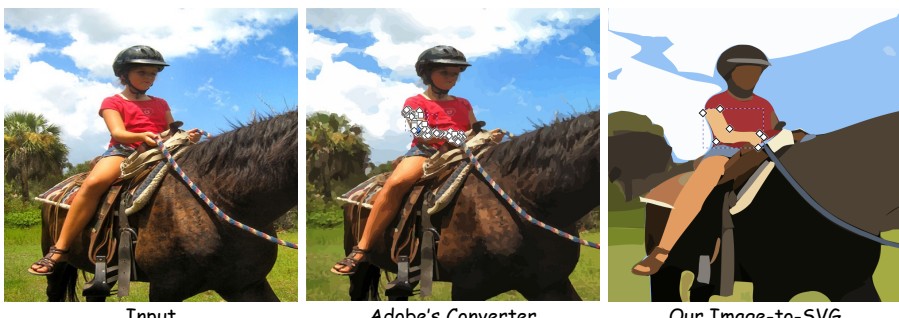

Input         Adobe's Converter         Our Image-to-SVG

Figure 11: Comparison between our Image-to-SVG (I2S) and Adobe's Image-to-SVG converter. Our I2S module is **explicitly optimized around semantic objects**, making it much **easier** for users to modify meaningful parts of the image. For example, if a user wants to adjust **the shape or position of an arm**, our SVG structure is significantly simpler than existing vectorization methods, which greatly facilitates intuitive editing.

should instead prioritize the physical realism enforced by the baseline model as shown in Figure 10. Users can flexibly adjust the guidance scale for different scenarios.

## A.5 COMPARE OUR IMAGE-TO-SVG WITH IMAGE VECTORIZATION METHODS

We provide several visual comparisons to highlight the differences between our SVG representation and existing image vectorization methods. Methods that optimize primarily for pixel fidelity can indeed preserve fine details accurately, but they often produce highly complex SVG curves, making user editing difficult. In contrast, our Image-to-SVG module is designed to prioritize semantic object structures, enabling much more intuitive and convenient modifications. For instance, as shown in Fig. 11, if a user wishes to adjust the shape or position of an object, such as a girl's arm, our SVG representation makes these edits significantly easier.

## A.6 CLARIFICATION OF LLM USAGE

In this work, we use LLMs to refine certain sentences in the paper by providing draft text and requesting suggestions on word choice and sentence structure.

