# OpenReview forum: "Vec2Pix: Controllable Image Synthesis via Semantic-aligned Vector Graphics"
_ICLR.cc/2026/Conference — ICLR 2026 Conference Desk Rejected Submission_

### Official Review · Reviewer_yAJS · 2025-10-23

**Soundness:** 2
**Presentation:** 4
**Contribution:** 2
**Rating:** 4
**Confidence:** 4

**Summary:**

The paper suggests a new controllable image editing pipeline - leveraging SVG to edit images using flow-matching based models.
SVG enables a layerwise, easy to control domain for editability.
The paper details a complete framework, transforming images to SVG and back to images, enabling a regeneration loop for user-friendly editing.

**Strengths:**

- The paper is mostly well-written, polished, and easy to follow.
- The qualitative results are impressive, and the suggested interface looks practical.
- The authors are willing to publish their dataset and code for future research.

**Weaknesses:**

- Limited Novelty - While the pipeline is effective, most of its components are already published, and the novelty of chaining them together is relatively modest
Specifically, the flexibility enabled by SVG is well known and already used to justify methods like DiffVG or LIVSS.
And, the components used for the image-to-SVG are based on prior works with limited novelty.
While the NPV module appears to suggest a novel alignment strategy, it's not well justified theoretically, and I'm unsure if this solution applies to any image-to-image frameworks or just SVG.
This also breaks the IID assumption of the Gaussian noise, as it might be spatially correlated (although supervised). A quantitative measure would be helpful. It might be that spatially correlated noise is beneficial for the editing task, but this would require more justification.

- Limited quantitative experiments -
SVG representation presents a lot more information than edges and depth.
A naive baseline, close to the suggested framework, should be included in the quantitative evaluation.
While acknowledging that the method is the first to suggest SVG-to-Image editing pipeline using diffusion, a baseline can be created to emphasize the novelty of the method.
For example, a non-layerwise, image-to-SVG method (like Adobe's or any other public method), and use an image-to-image editing framework?
In addition, it would be interesting to compare segmentation-guided diffusion editing to the suggested method, as it's a lot closer than edges or depth.

**Questions:**

- According to Figure 2, the VAE encoder's input is a rasterized SVG (image domain), while this is not entirely clear to me from the text. Does the encoder use a rendered image or work directly on the SVG bezier parameters?

---

> ### Author Response · Authors · 2025-11-21
> **Response to Reviewer yAJS**
>
> We sincerely thank Reviewer yAJS for your valuable comments. Please find the response below.
>
> > **W1.1: Novel components in Image-to-SVG (I2S)**
>
> The learning objectives of our I2S module are **fundamentally different from prior vectorization methods** such as LIVSS, O&R, DiffVG, and Bézier-Splatting. While **prior methods pursue pixel-level fidelity**, our objective is to obtain **simple, semantic-aligned, layer-wise, and truly editable SVGs within seconds**. To achieve such objective, we design the new I2S pipeline from such two aspects:
>
> (1) **Easy-to-edit SVG representation.**
>
> Prior fidelity-driven methods often generate overly complex SVGs. LIVSS, for example, initializes SVGs through a mask → polygon → SVG procedure, which frequently expands irregular boundaries into a large number of control points, producing SVGs that are difficult to edit. In contrast, our differentiable formulation directly optimizes a compact Bézier curve set with an explicit upper bound on control-point count, ensuring that the resulting SVGs stay simple and easy to edit. By constraining vector complexity from the beginning, our method produces clean, well-structured, and genuinely editable SVGs, making them practical for downstream editing tasks.
>
> (2) **Hierarchical vector parsing with high optimization efficiency.**
>
> LIVSS relies on a two-stage pipeline and a slow differentiable rasterizer, making it too inefficient for practical editing. In contrast, we jointly optimize a compact Bézier set in a single stage using both semantic masks and images for direct, stable supervision. By removing opacity blending (semantic regions are mutually exclusive) and enforcing no-overlap regularization within each layer, we ensure clean intra-layer boundaries between different elements and make the optimization objective easier. With our fast 2D-Gaussian rasterizer, vectorization converges in only a few dozen steps—over 7× faster than prior methods while producing higher-quality SVGs (Table 2).
>
>
>
>
> > **W1.2: I.I.D in Noise alignment (NPV)**
>
> The training of our **SVG-to-Image model in the first stage still follows the i.i.d. assumption** required by flow matching. **Our proposed NPV occurs only in the second satge**, built on top of the pretrained flow matching model.
>
> Our earlier experiments also confirmed that joint training leads to model collapse. In contrast, once a diffusion or flow-matching model has been pretrained, **the initial noise used to generate a specific image no longer needs to strictly satisfy the i.i.d. assumption**. **For example**, the initial noise obtained through **diffusion inversion** often contains spatial correlations, particularly in regions where the corresponding input image deviates from the natural-image manifold. Another example comes from **training-free image editing**. Several prior training-free methods [1–2] perform Image-to-Image (I2I) translation by modifying the initial noise or the intermediate noisy latents during the inference process of diffusion or flow models, verifying that the i.i.d. assumption is not required given a specific condtion image.
>
> Our S2I module aims to combine the strengths of such training-free I2I approaches (for example, initial noise rescheduling) and fine-tuning based I2I methods. Instead of computing the initial noise using a fixed hand-crafted formulation, we introduce a **second-stage training procedure that learns a predictable noise with both mean and variance conditioned on the SVG rendered images**. This allows the model to more effectively inject conditional information while preserving high fidelity in the generated results. In principle, the proposed noise alignment mechanism can be applied to a wide range of conditional generation or general editing tasks to improve the fidelity.
>
> Reference:
>
> [1] RF-Inversion: Semantic Image Inversion and Editing Using Recified Stochastic Differential Equations (ICLR'25)
>
> [2] SDEdit: Guided Image Synthesis and Editing with Stochastic Differential Equations (ICLR'22)

---

> ### Author Response · Authors · 2025-11-21
> **Response to Reviewer yAJS**
>
> > **W1.3: Effectiveness of NPV**
>
> The ablation study in Table 1 demonstrates the effectiveness of NPV on both reconstruction and editing quality.
> We also added more comparisons to integrate NPV into other conditional generation settings (e.g., Canny and Stroke). The results in the table below (with and without NPV) show that NPV generally improves reconstruction quality across different conditioning types. We further apply NPV to a general editing method (i.e., Flux-Knotext); because fidelity and generative diversity naturally form a trade-off in editing tasks, the resulting performance varies depending on the specific case. We provide examples in Fig. 9 in Appendix.
>
> > **Why NPV suitable for SVG-guided generation?** The noise-alignment module promotes better alignment between the generated content and the conditioning representations. Because our SVG representation captures the original image information more faithfully than sketches, masks, and other conditions (as shown by the reconstruction results in Table X), the NPV module is particularly well suited for SVG-conditioned generation, enabling stronger output alignment.
>
> > **W2.1: Quantitative results**
>
> We report quantitative results in Table 1 to evaluate both reconstruction and editing performance under different conditioning types, following the evaluation protocols adopted in prior conditional-generation work [Ominicontrol ICCV2025]. One advantage of our approach is that the SVG representation preserves richer information, providing an accurate and easy-to-edit conditioning representation that enables users to perform more precise manipulations.
>
> > **W2.2: More comparisons**
>
> Thanks for the suggestion. We have added the segmentation mask as an additional comparison in the table below. As mentioned earlier, the goal of our image-to-SVG (I2S) module differs from prior I2S methods, whose outputs are often difficult to modify. To further support this point, we include comparison between Adobe’s I2S converter and our I2S module as shown in Fig. 11 of the Appendix.
>
>
> | Representation        | Base Model | **Reconstruction** |        |        |        | **Editing** |        |        |        |
> |-----------------------|------------|--------------------|--------|--------|--------|-------------|--------|--------|--------|
> |                       |            | FID ↓             | LPIPS ↓ | SSIM ↑ | PSNR ↑ | FID ↓      | LPIPS ↓ | SSIM ↑ | PSNR ↑ |
> | Canny                 | FLUX.1     | 17.38              | 0.4582 | 0.42   | 12.28  | 22.48       | 0.5639 | 0.37   | 11.54  |
> | Canny w NPV                | FLUX.1     | 16.89              | 0.4501 | 0.44   | 12.65  | 21.76       | 0.5573 | 0.41   | 12.32  |
> | Depth                 | FLUX.1     | 19.65              | 0.5572 | 0.36   | 11.60  | 22.94       | 0.5932 | 0.36   | 11.37  |
> | Stroke                | FLUX.1     | 22.42              | 0.4721 | 0.45   | 17.13  | 25.35       | 0.5249 | 0.44   | 14.87  |
> | Stroke w NPV                | FLUX.1     | 21.88              | 0.4682 | 0.47   | 17.33  | 25.01       | 0.5204 | 0.46   | 15.13  |
> | Segmentation Mask                | FLUX.1     |      17.75         | 0.4513 |0.43 | 15.78 | 21.97| 0.5382 |0.43|14.34|
> | **Ours (SVG) w/o NPV** | **FLUX.1** | **16.03**          | **0.3687** | **0.47** | **17.42** | **18.87** | **0.3901** | **0.47** | **17.25** |
> | **Ours (SVG) w NPV**   | **FLUX.1** | **15.52**          | **0.3515** | **0.49** | **17.77** | **17.84** | **0.3836** | **0.50** | **17.72** |
>
> > **Q1: Rendered images?**
>
> Thanks for indicating the unclear point. The input condition is rendered images of SVG. We have fixed the unclear definition in Fig. 2.

---

### Official Review · Reviewer_qeYi · 2025-10-26

**Soundness:** 3
**Presentation:** 3
**Contribution:** 3
**Rating:** 6
**Confidence:** 4

**Summary:**

This paper presents a method that generate images with the condition of vector graphics. Meanwhile, the method perform vectorization of the generated images to enable further editings. The results demonstrated in the paper shows that the editing is effective for rough editing on the curve primitives. However, the undesired changes at the unedited regions make the proposed method suboptimal for image editing task. This issue makes me hesitate of giving higher ratings.

**Strengths:**

- The control through vector graphics primitive is straightforward and makes sense.
- The closed loop of generation and vectorization is useful.
- Multiple useful applications and scenarios are demonstrated.

**Weaknesses:**

- The proposed method seems like a combination of many existing things. This makes the overall method less novel. For example, the vectorization part is mainly based on LIVSS. And I am not really sure whether the vector-guided image generation part is novel. The condition, i.e., the rough vectorized image, is similar to a segmentation map which has been used for guided generation using ControlNet like method. I am not satisfied with the experiment without the discussion related to this aspect.
- It is very unclear what exactly the condition is and not clearly described in the paper. Is the rendered layer images? Or the curve parameters?
- The demonstrated results often introduce undesired visual artifacts, e.g., additional blurs of the girl's hair (Fig. 3) and the change of identity of the house and the shape of the rocks at river shore in Fig. 3. These parts are not the edited region but still got affected, which is not desired. This is critical for image editing task.

**Questions:**

- The overall implementation seems quite complicated because there are many steps. Do the authors consider to release the source code? Meanwhile, I think the Bezier splatting code is not released as well, therefore the implmenetation of the proposed method become more difficult.
- Why not trying to introduce some preservation strategies for the regions outside the editing region during the image editing? I think this is critical for image editing. The lack of discussing about this matter makes me feel that the proposed method is not suitable for image editing yet.

---

> ### Author Response · Authors · 2025-11-21
> **Response to Reviewer qeYi**
>
> We sincerely thank Reviewer qeYi for your valuable comments. Please find the response below.
>
> > **W1.1: Novelty of pipeline**
>
> * Image-to-SVG (I2S):
>
> The learning objectives of our I2S module are **fundamentally different from prior vectorization methods** such as LIVSS, O&R, DiffVG, and Bézier-Splatting. While **prior methods pursue pixel-level fidelity**, our objective is to obtain **simple, semantic-aligned, layer-wise, and truly editable SVGs within seconds**. To achieve such objective, we design the new I2S pipeline from such two aspects:
>
> (1) **Easy-to-edit SVG representation.**
>
> Prior fidelity-driven methods often generate overly complex SVGs. LIVSS, for example, initializes SVGs through a mask → polygon → SVG procedure, which frequently expands irregular boundaries into a large number of control points, producing SVGs that are difficult to edit. In contrast, our differentiable formulation directly optimizes a compact Bézier curve set with an explicit upper bound on control-point count, ensuring that the resulting SVGs stay simple and easy to edit. By constraining vector complexity from the beginning, our method produces clean, well-structured, and genuinely editable SVGs, making them practical for downstream editing tasks.
>
> (2) **Hierarchical vector parsing with high optimization efficiency.**
>
> LIVSS relies on a two-stage pipeline and a slow differentiable rasterizer, making it too inefficient for practical editing. In contrast, we jointly optimize a compact Bézier set in a single stage using both semantic masks and images for direct, stable supervision. By removing opacity blending (semantic regions are mutually exclusive) and enforcing no-overlap regularization within each layer, we ensure clean intra-layer boundaries between different elements and make the optimization objective easier. With our fast 2D-Gaussian rasterizer, vectorization converges in only a few dozen steps—over 7× faster than prior methods while producing higher-quality SVGs (Table 2).
>
>
>
>
> * SVG-to-Image (S2I): We designed a **NEW** conditional generation framework for S2I; **NOT direct controlnet-style**.
>
> (1) Since our SVG representation preserves richer and more accurate structural information than prior conditions (e.g., sketch or depth), we introduce a dedicated **Noise Prediction from Vector (NPV) module and two-stage training strategy** to better align the SVG conditions with the generated results—particularly in terms of **color fidelity and boundary geometry**. This leads to noticeably improved reconstruction quality, as shown in Table 1 and Figure 6(a) (w and w/o NPV).
>
> (2) Our S2I module aims to combine the strengths of both training-free Image-to-Image (I2I) approaches (for example, initial noise rescheduling [1–2]) and fine-tuning based I2I methods. Instead of computing the initial noise using a fixed training-free formulation, **we introduce a NPV module that learns a predictable initial noise with both mean and variance conditioned on the SVG rendered images.** This allows the model to more effectively inject conditional information while preserving high fidelity in the generated results. In principle, the proposed NPV can be applied to a wide range of conditional generation or general editing tasks.
>
> Reference:
>
> [1] RF-Inversion: Semantic Image Inversion and Editing Using Recified Stochastic Differential Equations (ICLR'25)
>
> [2] SDEdit: Guided Image Synthesis and Editing with Stochastic Differential Equations (ICLR'22)

---

> ### Author Response · Authors · 2025-11-21
> **Response to Reviewer qeYi**
>
> > **W1.2: Compared to Segmentation Mask-guided Controlnet**
>
> We added the segmentation mask as an additional condition for comparison using the OminiControl (ICCV’25) framework for re-training. The results in the table below further validate the effectiveness of our SVG representation.
>
> | Representation        | Base Model | **Reconstruction** |        |        |        | **Editing** |        |        |        |
> |-----------------------|------------|--------------------|--------|--------|--------|-------------|--------|--------|--------|
> |                       |            | FID ↓             | LPIPS ↓ | SSIM ↑ | PSNR ↑ | FID ↓      | LPIPS ↓ | SSIM ↑ | PSNR ↑ |
> | Canny                 | FLUX.1     | 17.38              | 0.4582 | 0.42   | 12.28  | 22.48       | 0.5639 | 0.37   | 11.54  |
> | Canny w NPV                | FLUX.1     | 16.89              | 0.4501 | 0.44   | 12.65  | 21.76       | 0.5573 | 0.41   | 12.32  |
> | Depth                 | FLUX.1     | 19.65              | 0.5572 | 0.36   | 11.60  | 22.94       | 0.5932 | 0.36   | 11.37  |
> | Stroke                | FLUX.1     | 22.42              | 0.4721 | 0.45   | 17.13  | 25.35       | 0.5249 | 0.44   | 14.87  |
> | Stroke w NPV                | FLUX.1     | 21.88              | 0.4682 | 0.47   | 17.33  | 25.01       | 0.5204 | 0.46   | 15.13  |
> | Segmentation Mask                | FLUX.1     |      17.75         | 0.4513 |0.43 | 15.78 | 21.97| 0.5382 |0.43|14.34|
> | **Ours (SVG) w/o NPV** | **FLUX.1** | **16.03**          | **0.3687** | **0.47** | **17.42** | **18.87** | **0.3901** | **0.47** | **17.25** |
> | **Ours (SVG) w NPV**   | **FLUX.1** | **15.52**          | **0.3515** | **0.49** | **17.77** | **17.84** | **0.3836** | **0.50** | **17.72** |
>
> > **W2: Condition format**
>
> Thanks for indicating the unclear point. The input condition is rendered images of SVG. We have fixed the unclear definition in Fig. 2.
>
> > **W3: May Background change & Q2: Region preservation**
>
> * The background consistency can also be further improved using the **explicit mask**. **The SVG formulation naturally provides explicit curve-wise control signals**: once a user modifies a Bézier curve, we can directly derive a corresponding binary mask that precisely indicates the edited region. By integrating this explicit mask into the generation process, we can further enhance background and global consistency while keeping changes localized to the user-specified areas. **We updated the first and second examples with mask in Fig. 3 achieving better consistency** and we will provide an option using explicit mask in our system. (Most of our other results also demonstrate good consistency, as shown in Fig. 4 and the examples on our homepage.)
> * Maintaining perfect consistency during image editing remains a challenging open problem. This difficulty persists across many existing methods, especially for large editing region cases. As shown in Fig. 4, even state-of-the-art open-source and commercial models often struggle to preserve fundamental attributes—such as character identity or hair texture—across edited results. In contrast, our method a achieves stronger consistency especially with derived mask, largely due to the rich information encoded in our SVG representation.
>
> > **Q1: Many steps of pipeline and code release?**
>
> Some steps in Figure 2 are optional. For example, when an existing SVG is available, users can directly modify it, requiring only Steps 1–2. Both our I2S and S2I modules are highly efficient: as reported in the main paper, the Image-to-SVG (I2S) stage takes around 3 seconds and the SVG-to-Image (S2I) stage around 14 seconds (A100 GPU), adding only a small overhead relative to the backbone (i.e., ~14s for the inference of Flux). In contrast, recent systems such as GPT-4o and Qwen-Image-Edit typically require several minutes.
>
> We will release all codes, benchmark, and datasets.

---

### Official Review · Reviewer_XUMh · 2025-10-31

**Soundness:** 3
**Presentation:** 2
**Contribution:** 2
**Rating:** 2
**Confidence:** 4

**Summary:**

This paper proposes a novel framework for element-level controllable image generation by leveraging simplified vector graphics (VGs) as the conditioning representation. Instead of traditional control signals such as sketches, layouts, or depth maps, the method converts images to hierarchical SVG-like vector structures and enables users to edit shapes, colors, and object components directly. A bidirectional loop is introduced, consisting of SVG-guided image synthesis and image-to-SVG parsing via differentiable vectorization and B-spline rendering. A noise-prediction module aligns vector structures with diffusion sampling. Experiments show effective object-level manipulation, layout adjustment, and fine-grained editing across diverse tasks.

**Strengths:**

1. New controllable generation paradigm using vector graphics for semantic and geometric control, beyond typical spatial conditioning (layout/depth/sketch).

2. Closed SVG⇌image loop enabling iterative refinement and editable structure, bridging design workflows and generative models.

3. Strong technical system: vector parsing, differentiable rasterization, noise alignment, LoRA adapters, and multimodal diffusion integration.

**Weaknesses:**

1. Limited technical novelty. The core method primarily combines an existing hierarchical vector parsing pipeline LIVSS and O&R [2] with a ControlNet-style conditional guidance setup. The overall architecture closely parallels prior works such as Densediffusion [1], making the contribution appear incremental and largely engineering-oriented rather than introducing fundamentally new modeling innovations.

2. Vectorization quality and practical usability remain weak. The hierarchical vector graphics obtained by the proposed pipeline are still coarse and far from production-ready. The editable control via point dragging is highly constrained in practice, offering only limited manipulation capability for complex shapes and semantic elements. As such, the real usability of the system does not fully match the claimed level of controllability.

3. Efficiency and scalability concerns. The system depends on differentiable rendering and per-shape optimization, leading to slow inference and interaction cycles. This latency considerably limits scalability to high-resolution or complex scenes

4. No clear advantage over existing image editing workflows. Compared to modern image editing pipelines (e.g., prompting-based iterative editing, sketch-guided refinement, or mask-based diffusion workflows), the proposed system does not show a clear qualitative advantage. In many cases, the generated results appear similar or even inferior to existing diffusion-based editing methods, weakening the practical motivation.

Reference:
1. Dense Text-to-Image Generation with Attention Modulation
2. Optimize and Reduce: A Top-Down Approach for Image Vectorization

**Questions:**

1. Figure 2 and the method description are confusing. Is the condition input to the diffusion model the SVG file itself (e.g., text, paths), or the rasterized pixel image derived from the SVG?

2. What is the backbone of the Flow model shown in Figure 2? If it is DiT, the diagram should not depict a U-Net-style architecture, as this is misleading.

---

> ### Author Response · Authors · 2025-11-21
> **Response to Reviewer XUMh**
>
> We sincerely thank Reviewer XUMh for your valuable comments. Please find the response below.
>
> > **W1.1: Technical novelty**
>
> * Image-to-SVG (I2S):
>
> The learning objectives of our I2S module are **fundamentally different from prior vectorization methods** such as LIVSS, O&R, DiffVG, and Bézier-Splatting. While **prior methods pursue pixel-level fidelity**, our objective is to obtain **simple, semantic-aligned, layer-wise, and truly editable SVGs within seconds**. To achieve such objective, we design the new I2S pipeline from such two aspects:
>
> (1) **Easy-to-edit SVG representation.**
>
> Prior fidelity-driven methods often generate overly complex SVGs. LIVSS, for example, initializes SVGs through a mask → polygon → SVG procedure, which frequently expands irregular boundaries into a large number of control points, producing SVGs that are difficult to edit. In contrast, our differentiable formulation directly optimizes a compact Bézier curve set with an explicit upper bound on control-point count, ensuring that the resulting SVGs stay simple and easy to edit. By constraining vector complexity from the beginning, our method produces clean, well-structured, and genuinely editable SVGs, making them practical for downstream editing tasks.
>
> (2) **Hierarchical vector parsing with high optimization efficiency.**
>
> LIVSS relies on a two-stage pipeline and a slow differentiable rasterizer, making it too inefficient for practical editing. In contrast, we jointly optimize a compact Bézier set in a single stage using both semantic masks and images for direct, stable supervision. By removing opacity blending (semantic regions are mutually exclusive) and enforcing no-overlap regularization within each layer, we ensure clean intra-layer boundaries between different elements and make the optimization objective easier. With our fast 2D-Gaussian rasterizer, vectorization converges in only a few dozen steps—over 7× faster than prior methods while producing higher-quality SVGs (Table 2).
>
>
>
>
> * SVG-to-Image (S2I): We designed a **NEW** conditional generation framework for S2I; **NOT direct controlnet-style**.
>
> (1) Since our SVG representation preserves richer and more accurate structural information than prior conditions (e.g., sketch or depth), we introduce a dedicated **Noise Prediction from Vector (NPV) module and two-stage training strategy** to better align the SVG conditions with the generated results—particularly in terms of **color fidelity and boundary geometry**. This leads to noticeably improved reconstruction quality, as shown in Table 1 and Figure 6(a) (w and w/o NPV).
>
> (2) Our S2I module aims to combine the strengths of both training-free Image-to-Image (I2I) approaches (for example, initial noise rescheduling [1–2]) and fine-tuning based I2I methods. Instead of computing the initial noise using a fixed training-free formulation, **we introduce a NPV module that learns a predictable initial noise with both mean and variance conditioned on the SVG rendered images.** This allows the model to more effectively inject conditional information while preserving high fidelity in the generated results. In principle, the proposed NPV can be applied to a wide range of conditional generation or general editing tasks.
>
>
> Reference:
>
> [1] RF-Inversion: Semantic Image Inversion and Editing Using Recified Stochastic Differential Equations (ICLR'25)
>
> [2] SDEdit: Guided Image Synthesis and Editing with Stochastic Differential Equations (ICLR'22)

---

> ### Author Response · Authors · 2025-11-21
> **Response to Reviewer XUMh**
>
> > **W1.2: Compared to DenseDiffusion**
>
> (1) Our method differs fundamentally from DenseDiffusion, which does **NOT address editability or fine-grained controllability**. Its segmentation-mask conditioning does not form an editable representation, whereas our semantic SVG formulation enables structured, object-level manipulation that DenseDiffusion cannot support.
>
> (2) Segmentation masks in DenseDiffusion are inherently coarse and contain far less structural information than our semantic SVG representation. By preserving object-level geometry and attributes, our SVG formulation allows more accurate reconstruction and thus enables high-fidelity, object-wise editing beyond the capabilities of DenseDiffusion. We also **added comparison with segmentation mask** in below table. Our SVG representations have obvious strengths on both reconstruction and editing quality.
>
> | Representation        | Base Model | **Reconstruction** |        |        |        | **Editing** |        |        |        |
> |-----------------------|------------|--------------------|--------|--------|--------|-------------|--------|--------|--------|
> |                       |            | FID ↓             | LPIPS ↓ | SSIM ↑ | PSNR ↑ | FID ↓      | LPIPS ↓ | SSIM ↑ | PSNR ↑ |
> | Canny                 | FLUX.1     | 17.38              | 0.4582 | 0.42   | 12.28  | 22.48       | 0.5639 | 0.37   | 11.54  |
> | Depth                 | FLUX.1     | 19.65              | 0.5572 | 0.36   | 11.60  | 22.94       | 0.5932 | 0.36   | 11.37  |
> | Stroke                | FLUX.1     | 22.42              | 0.4721 | 0.45   | 17.13  | 25.35       | 0.5249 | 0.44   | 14.87  |
> | Segmentation Mask                | FLUX.1     |      17.75         | 0.4513 |0.43 | 15.78 | 21.97| 0.5382 |0.43|14.34|
> | **Ours (SVG) w/o NPV** | **FLUX.1** | **16.03**          | **0.3687** | **0.47** | **17.42** | **18.87** | **0.3901** | **0.47** | **17.25** |
> | **Ours (SVG) w NPV**   | **FLUX.1** | **15.52**          | **0.3515** | **0.49** | **17.77** | **17.84** | **0.3836** | **0.50** | **17.72** |
>
> > **W2: SVG usability**
>
> * SVG still coarse?
>
> Our goal is not to reproduce pixel-level illustration details, but to provide a semantic-object–level SVG representation that enables intuitive, fine-grained editing and user control, capabilities that existing conditional generation or editing approaches cannot offer.
>
> * Limited manipulation capability?
>
> Natural images offer the finest level of detail, but they are difficult to be directly edited, which presents a trade-off. Our work achieves a good balance between these aspects, which **decomposes an image/scene into a structured set of semantic objects, each represented with its own vector primitives that can be flexibly recomposed and modified** (comparisons refer to Fig. 11 in Appendix).
> For cases that require finer-grained edits, our system also provides **multiple layers of SVG representations**, allowing users to select the appropriate level of vector-graphic complexity. Moreover, since our Image-to-SVG (I2S) module is differentiable, users can adjust its parameter scales to obtain SVGs with varying degrees of complexity. We also illustrate several fine-grained control examples, such as finger details in Fig. 5.
>
> > **W3: Efficiency**
>
> As reported in the main paper, the Image-to-SVG (I2S) stage takes ~3 seconds and the SVG-to-Image (S2I) stage ~14 seconds, introducing only a small overhead relative to the backbone (i.e., ~14s of Flux). In contrast, recent systems such as GPT-4o or Qwen-Image-Edit typically require over two minutes.
>
> > **W4: Advantages over existing editing**
>
> (1) Compared with text-prompt–based editing, our method enables precise control over object positions and attributes, offering significant advantages for manipulating object locations and shapes, especially scenes with multiple objects.
>
> (2) Compared with mask-guided and sketch-guided approaches, our representation is more accurate and provides a direct SVG-based editing interface that allows users to precisely adjust object shapes and positions.
>
> (3) Additionally, our hierarchical representation offers substantial benefits for object removal and replacement, as it helps maintain background consistency after editing.
>
> > **Q1: Condition input**
>
> Thanks for indicating the unclear point. The input condition is rendered images of SVG. We have fixed the unclear definition in Fig. 2.
>
> > **Q2: Backbone in Fig 2**
>
> We use Flux as our backbone, which follows the DiT architecture. We have also revised Fig. 2 to be more general in order to avoid potential misunderstandings.

---

> ### Comment · Reviewer_XUMh · 2025-11-27
> **Thank you for the response and the detailed discussion. I Maintain Original Assessment**
>
> 1. I appreciate the authors’ detailed rebuttal, but it does not resolve my primary concern regarding the claimed advantages of SVG-based vectorization. Throughout the paper and the response, the authors emphasize that their method produces structurally clean, editable, and efficient SVGs. However, if the paper intends to claim advantages in vectorization itself, then it must include systematic comparisons against existing hierarchical vectorization pipelines in terms of reconstruction fidelity, structural detail, efficiency, and user preference. The rebuttal simultaneously argues that the method does not pursue pixel-level fidelity and focuses only on providing a semantic, layered representation. This position contradicts the earlier claims of superiority and leaves a fundamental inconsistency unaddressed: if the goal is not high-fidelity vectorization, then the claimed advantages are overstated; if the goal is to improve vectorization, then the comparisons provided are insufficient.
>
> 2. Another unresolved issue concerns the actual conditioning mechanism of the proposed model. The authors clarify that the diffusion model does not ingest SVG structures directly but instead receives rasterized images rendered from those SVGs. This makes the conditioning pipeline structurally equivalent to DenseDiffusion and other raster-conditioned diffusion models, fundamentally relying on pixel-space guidance rather than vector-level conditioning. As a result, the distinction the authors draw between their approach and DenseDiffusion is not meaningful at the modeling level. In practice, the SVG representation functions more as a user-interface abstraction than as a fundamentally different conditioning signal, and the rebuttal does not justify the claim that the proposed framework constitutes a new or substantially different generative paradigm.
>
> 3. The rebuttal repeatedly asserts that DenseDiffusion “cannot edit,” which is inaccurate. DenseDiffusion supports brush-based mask creation, and when masks are drawn using colors that follow object boundaries, the editing experience can be comparable to dragging vector control points for structural adjustments. Each method possesses different interaction affordances, and neither is inherently superior across all editing scenarios. The SVG-based manipulation described in the paper—primarily selection and dragging—offers only limited advantages and does not clearly surpass mask-based or sketch-based editing in expressive power or practicality, particularly for complex shapes or natural images. Therefore, the authors’ response does not sufficiently address my concerns regarding the claimed usability benefits, nor does it clarify the real functional gap between the proposed system and established diffusion-based editing workflows. For these reasons, I maintain my original evaluation.
>
> 4. In addition, I remain unconvinced by the claimed advantages of the proposed hierarchical vector representation for interactive editing. The paper states that images are decomposed into layered SVG structures, yet in practice an individual object is often composed of multiple vector primitives distributed across several layers. This raises a fundamental usability question: how does the system reliably select an entire object for manipulation, such as dragging or repositioning, when its components are fragmented across multiple Bézier curves or sublayers? Such grouping is non-trivial and typically requires an explicit object-level aggregation mechanism; without it, users would need to manually group all constituent elements, which is itself a heavy and error-prone interaction. The rebuttal does not explain how this issue is handled, nor does it demonstrate that the proposed SVG workflow offers a practical advantage over existing mask- or region-based editing tools in this respect.

---

> ### Author Response · Authors · 2025-11-28
>
> We respectfully disagree with the comments and hope the following clarifications address your concerns.
>
> 1. Our paper does NOT claim advantages in vectorization quality for reconstruction. Our contributions focus only on efficiently producing semantic-aligned and editable SVGs for downstream SVG-to-image synthesis and editing, rather than outperforming reconstruction-oriented image vectorization pipelines. Therefore, comparisons to DiffVG, Bézier-Splatting, or other reconstruction-focused image vectorization methods towards reconstruction accuracy are not aligned with our problem setting, as these approaches do not generate semantic-aligned SVGs and are not applicable to semantic editing. For the only relevant semantic-aligned baseline, LIVSS, we already include quantitative comparisons in Table 2, where our method achieves slightly **better PSNR** and is **significantly faster**. In addition, the appendix (Fig. 8) provides extensive qualitative results demonstrating that our method indeed produces clean, semantic-aligned SVGs.
>
> 2. **Sharing the same input/output space with prior is not a reason for rejection. For example, for image editing, style transfer, nearly all methods take image as both input and output**—yet they introduce fundamentally different modeling ideas and remain fully valid and publishable contributions. The domain of inputs and outputs alone does not define novelty.
> Our method introduces both novel and effective innovations: We introduce a **NEW** conditional generation framework with a two-stage noise-prediction from vector (NPV) design. We demonstrate NPV **not only benefits SVG-to-image generation but also yields clear improvements in image editing tasks** (Fig. 9 and Table 3 of Appendix). Moreover, the core idea of our noise prediction strategy is generalizable and can, in principle, be applied to raster-conditioned models such as DenseDiffusion.
>
> 3. As you mentioned, every type of representation has its own advantages, including SVG. Many recent works have also explored SVG-related research, demonstrating their practical usability. Therefore, its “usability benefits” should not be questioned.
> Our Vec2Pix is a **different type of interaction** compared to existing representations and **provides precise shape manipulation and accurate localization**, which is particularly advantageous in structural adjustments. Besides, **SVG stores more information, leading to better editing consistency**. We also include comparisons with segmentation-mask editing to support this point. SVG objects can be moved, reshaped, resized, reordered, or replaced while preserving clean boundaries. In contrast, while brush-based mask editing can produce the desired shape, it often requires **considerable manual effort**. In real applications, users often need to redraw the mask many times to achieve a satisfactory shape, **making repeated erasing and repainting** far more cumbersome than **simply adjusting a few control points** in our SVG representation.
>
> 4. We respectfully clarify that the reviewer’s concern arises from a misunderstanding of our representation. Fragmentation of vector primitives indeed occurs in reconstructive vectorization pipelines such as DiffVG or Bézier-Splatting, where primitives are optimized independently and may unintentionally split an object across multiple SVGs, making grouping non-trivial. Our method does not follow this paradigm.
> We first obtain layer-wise semantic masks using diffusion + SAM, and then construct an **explicit hierarchical semantic tree** by organizing these regions according to their semantic relationships and relative mask areas. This semantic tree provides a fixed, object-level grouping prior: during the subsequent vector optimization, the hierarchy remains unchanged, and only the primitives within each node are refined. If grouping is required, it is handled directly by the semantic tree, which automatically aggregates all Bézier primitives under the selected node. As shown in Fig. 11 of Appendix, our SVG optimization operates at an object level, allowing users to manipulate meaningful parts (e.g., arms, hats). The multi-layered SVGs in system to further enable choosing an appropriate abstraction level for practical editing workflows.

---

### Official Review · Reviewer_GJ3f · 2025-11-01

**Soundness:** 3
**Presentation:** 2
**Contribution:** 3
**Rating:** 6
**Confidence:** 5

**Summary:**

This paper proposes to tackle the core challenge in image generation: achieving fine-grained, intuitive control over image elements. It proposes a novel framework for layer-wise controllable generation using simplified, semantic-aligned vector graphics as an intermediate representation. By leveraging this intermediate representation within a diffusion-based generative pipeline, Vec2Pix enables element-level control over image synthesis. Experiments demonstrate its effectiveness across various applications, including image editing, object-level manipulation, and fine-grained content creation.

**Strengths:**

1. The idea of utilizing vector graphics as the control medium is novel and interesting. VGs are inherently object-based, hierarchical, and easily editable; the use of layer-wise initialization further facilitates this control paradigm.
2. The proposed method enables intuitive manipulation operations that are highly relevant for creative workflows, interactive design tools, and content creation applications.
3. The VG initialization achieves efficient and effective reconstructive results, and the method overall achieves good results for controllable image generation.

**Weaknesses:**

1. Trade-off between VG complexity and fidelity: VGs have inherent limitations in representing photorealistic details. A highly complex scene (e.g., water, smoke) might require an extremely complex VG, which could be difficult to edit and inefficient to synthesize from. The paper needs to clarify how its "simplified VG" representation balances editability with the ability to guide high-fidelity image generation.
2. The paper lacks clarity on how SVG vector graphics are actually represented and processed in the pipeline. Fig. 2 shows the VAE taking "SVG" as input, but it is unclear whether this refers to (a) tokenized SVG code/markup, or (b) rendered raster images of the SVG. If the method uses rendered images, then the "SVG" should be "SVG's rendered image".
3. It seems that the method does not guarantee editing consistency, particularly with respect to background preservation. As shown in Fig. 3, when objects are added or removed, the background undergoes changes that are unrelated to the editing operation.

**Questions:**

1. How sensitive is the method to the quality of the initial vector parsing? How is "semantic alignment" defined and evaluated?
2. "Seamlessness" of Edits: When a user modifies a VG element, how does the model handle the complex interactions between that element and the rest of the image (e.g., lighting, reflections, shadows, occlusion)? The abstract claims this is "seamless" and "photorealistic" but this is one of the most difficult problems in controllable synthesis. If handled poorly, edited objects will look "pasted on" and inconsistent with the background (OmniPaint [1], Gen-omnimatte [2]).

[1] OmniPaint: Mastering Object-Oriented Editing via Disentangled Insertion-Removal Inpainting ICCV'25

[2] Generative Omnimatte: Learning to Decompose Video into Layers CVPR'25

---

> ### Author Response · Authors · 2025-11-21
> **Response to Reviewer GJ3f**
>
> We sincerely thank Reviewer GJ3f for your valuable comments. Please find the response below.
>
> > **W1: Tradeoff between VG complexity and fidelity**
>
> The complexity–fidelity tradeoff is **a common challenge** in conditional generation, such as generation conditioned on sketches or pose skeletons. In contrast, our pipeline decomposes an image/scene into a structured set of semantic objects, each represented with its own vector primitives that can be flexibly recomposed and modified.
>
> In our Vec2Pix system, we provide **multiple layers of SVG representations** for users to choose from, each corresponding to a different level of vector-graphic complexity. Since our I2S module is differentiable, users can also adjust its parameter scales to obtain SVGs with different levels of complexity. This design aligns with the goal of our Image-to-SVG (I2S) module, which generates **efficient, multi-layer, and user-friendly** vector representations that support flexible editing. The appropriate VG complexity depends on the task and the level of detail a user wishes to modify. However, for all examples shown in the paper, we adopt the same SVG layer, which already proves sufficient for most scenarios and supports effective semantic-level object control and manipulation.
>
> > **W2: How SVG processed in the pipeline?**
>
> Thanks for indicating the unclear point. The input condition is rendered images of SVG. We have fixed the unclear definition in Fig. 2.
>
> > **W3: Editing consistency**
> * The background consistency can also be further improved using the **explicit mask**. **The SVG formulation naturally provides explicit curve-wise control signals**: once a user modifies a Bézier curve, we can directly derive a corresponding binary mask that precisely indicates the edited region. By integrating this explicit mask into the generation process, we can further enhance background and global consistency while keeping changes localized to the user-specified areas. **We updated the first and second examples with mask in Fig. 3 achieving better consistency** and we will provide an option using explicit mask in our system. (Most of our other results also demonstrate good consistency, as shown in Fig. 4 and the examples on our homepage.)
> * Maintaining perfect consistency during image editing remains a challenging open problem. This difficulty persists across many existing methods, especially for large editing region cases. As shown in Fig. 4, even state-of-the-art open-source and commercial models often struggle to preserve fundamental attributes—such as character identity or hair texture—across edited results. In contrast, our method a achieves stronger consistency especially with derived mask, largely due to the rich information encoded in our SVG representation.
>
> > **Q1: How sensitive of initial vector parsing and semantic alignment?**
>
> Our initial vector parsing is far less sensitive compared to methods like DiffVG or Bézier-Splatting because: (1) we initialize vector primitives from SAM’s semantic masks, which are deterministic and produce stable one-to-one mappings to SVG elements; (2) the layer-wise structure keeps masks within each layer minimally overlapping, and their clean semantic boundaries reduce ambiguity and stabilize optimization; and (3) we refine vectors for only ~50 steps to align with semantic masks and image supervision, avoiding the accumulated randomness of long optimization loops (e.g., 10,000 steps in Bézier-Splatting).
>
> The semantic alignment in our Image-to-SVG (I2S) module, enabled by SAM-based initialization and our non-overlap optimization, produces semantic-object-level parsing and non-overlapping SVG curves that greatly improve editing usability. We provide a comparison in Fig. 11 (Appendix) to highlight the differences between our I2S and classic I2S methods.
>
> > **Q2: Seamlessness of editing**
>
> We employ the guidance scale factor to control the alignment between the input conditions and the generated results. This factor also governs how the edited object interacts with its surrounding environment. Specifically, it **determines whether the generation should adhere more strictly to the input SVG or prioritize the physical realism imposed by the baseline model**. In the latter case, the surrounding shadows and reflections are generated accordingly, while only the object itself is edited. Examples are provided in Fig. 10 of the Appendix.
>
> In practice, users do not need to edit complex SVG representations to adjust shadows or reflectance. Even simplified SVG inputs are sufficient to edit complex scenes, and the corresponding natural environmental effects can be generated automatically.

---

### Author Response · Authors · 2025-11-21
**Revision Summary**

We sincerely thank all reviewers for their valuable comments and constructive suggestions.

## Revision Summary

* Added additional comparisons on segmentation masks, demonstrating that our SVGs are editable and exhibit improved color and structure (Table 1).
* Added an option to use an explicit mask to further enhance background consistency; this mask can be easily derived from the edited SVG and the original one (updated results in Fig. 3).
* Added more comparisons for Noise Prediction from Vector (NPV), including results across different conditions and general editing models (Table 3 and Fig. 9).
* Added a comparison between our Image-to-SVG module and Adobe’s converter in Fig. 11, illustrating how our semantic-object-level SVG representation enables much easier user editing.
* Added an ablation study on different condition-guidance scales in Fig. 10, which enables controlling whether the generation should adhere more strictly to the input SVG or prioritize the physical realism imposed by the baseline model.
* Clarified the previously unclear definition in Fig. 2.

## Code Release

We will release all codes, benchmark, and datasets.

---

### Note · Program_Chairs · 2026-01-17
**Submission Desk Rejected by Program Chairs**

The following references in this submission do not refer to real documents and/or have major errors in bibliographic information:

 Mengqi Zhang, Xiaoyang Wang, Ping Luo, Dong Yan, and Lei Wang. Text-guided image editing with diffusion models. In CVPR, 2023b.
Runbin Liu, Chengyue Gong, Shitong Xu, and Furong Huang. Flow straight and fast: Learning to generate and transfer data with rectified flow. In International Conference on Learning Representations (ICLR), 2023.
Yuxuan Cao, Zixuan Chen, Dongdong Wang, and Changsheng Xu. Ssr-encoder: Identity-preserving encoder for image-conditioned diffusion models. In arXiv preprint arXiv:2312.10938, 2023.
Bowen Wang, Timo Schick, Henning Schäfer, Edward J Hu, Mike Lewis, Luke Zettlemoyer, Xiangning Chen, Omer Levy, and Mohit Bansal. Instructpix2pix: Learning to follow image editing instructions. In CVPR, 2023.